# Simple Context Compression: Mean-Pooling and Multi-Ratio Training

## Abstract

A common strategy to reduce the computational costs of using long contexts in retrieval-augmented generation (RAG) with large language models (LLMs) is *soft context compression*, where the input sequence is transformed into a shorter continuous representation. We develop a lightweight and simple mean-pooling approach that consistently outperforms the widely used compression-tokens architecture, and study training the same compressor to output multiple compression ratios. We conduct extensive experiments across in-domain and out-of-domain QA datasets, as well as across model families, scales, and compression ratios. Overall, our simple mean-pooling approach achieves the strongest performance, with a relatively small drop when training for multiple compression ratios. More broadly though, across architectures and training regimes the trade-offs are more nuanced, illustrating the complex landscape of compression methods.

## 1 Introduction

Reasoning over long documents is common in scenarios of retrieval-augmented generation (RAG). This is a computationally costly process, both as far as time and memory. Time is impacted by processing the document itself and self-attending over its computed representations in later parts of the generation process. Memory costs spike due to the key-value (KV) cache of the processed document. The common way to reduce these costs is *soft context compression*, where a sequence of continuous representations is pre-computed. This representation is compressed in the sense that its length is significantly lower than the document length, thereby reducing both time and memory costs of reasoning over the document. This compressed representation is computed once, and then retrieved as needed. This important problem is receiving significant attention (e.g., Ge et al., 2024; Cheng et al., 2024a; Dai et al., 2025).

We study both compressor model design and the compression training process, with simplicity in mind. The compression encoder we design is composed of an encoding LLM, and straightforward mean-pooling operations to collapse together representations to achieve a target compression ratio. This approach adds no parameters beyond the encoder LLM, and the computation beyond the encoding of the document is minimal. It also naturally allows for compression ratio flexibility, raising the question of the benefits or downsides of training the same compressor to support multiple compression ratios. We design a simple training objective and process to achieve this. This is motivated foremost by the benefit of training a single model to serve different compute budgets, rather than maintaining and training a separate model for each compression ratio. It also allows to examine if and when training to compress at multiple ratios can perform better than training for a single ratio.

We construct a rigorous evaluation suite using multiple question-answering (QA) datasets. We distinguish between datasets that are part of our training set, and these that are completely held-out, allowing us to better gauge generalization. We conduct a battery of experiments, across three model families, model scales, and variations of both compressor architecture and multi-ratio training.

We find that our approach consistently outperforms the conventional compression-tokens approach, while being more efficient. In addition, we also show that by simply altering the attention pattern in the conventional compression tokens method, one can mitigate the gap between the approaches significantly, albeit not entirely. Our multi-ratio training experiments reveal that it is possible to train and deploy only a single model for a wide range of compression ratios with only minor performance drops. Interestingly, our proposed enhancement for the compression-tokens approach even benefits

from multi-ratio training. A comparison of compression performances for scales between 0.6B to 8B shows that compression quality increases with scale, amplifying the benefits of applying such compression methods to larger models. Code, data, and models will be released upon publication.

## 2 TASK DEFINITION

Soft context compression is an approach where a document of length $L$ is mapped to a sequence of vectors of length $C$, where $L \gg C$. While the original document can be described as a sequence of tokens, the compression is made of dense continuous vectors, hence *soft*. This process allows an LLM that uses the compressed version of the document to invest significantly less computation, both in time and key-value (KV) cache space, both reduced from dependence on $L$ to dependence on $C$. This benefit increases with repeated use of the document, as is likely in RAG scenarios.

Formally, we define soft context compression to support multiple compression ratios. Let $\mathcal{M}$ be a language model and $\mathcal{R} \subseteq \mathbb{N}_+$ the admissible set of compression ratios. Let $\mathcal{V}$ denote the vocabulary and $d$ the embedding dimension of $\mathcal{M}$. The goal of learning is to construct a compression function

$$f_c : \mathcal{V}^L \times \mathcal{R} \to \mathbb{R}^{C \times d} \ , \tag{1}$$

which maps a token sequence $T = (t_1, \ldots, t_L)$, $t_i \in \mathcal{V}$ of length $L$ and a ratio $r \in \mathcal{R}$ to a compressed representation of $C$ vectors of dimension $d$. The length $C$ is determined by the specified ratio $C = \lceil L/r \rceil$.

An ideal compressor $f_c$ preserves the conditional distribution of the model using the compressed version for any prompt $P$:

$$p_{\mathcal{M}}(\cdot \mid T, P) \ \approx \ p_{\tilde{\mathcal{M}}}(\cdot \mid f_c(T; r), P) \ , \tag{2}$$

where $\tilde{\mathcal{M}}$ is an adapted version of $\mathcal{M}$, for example augmented with lightweight parameters such as LoRA (Hu et al., 2022) modules that can be fused into the base model without altering its capacity.

## 3 BACKGROUND AND RELATED WORK

**Soft Context Compression** A dominant line of research on context compression adopts the use of artificial *compression tokens*. As shown in Figure 1b, a sequence of length $L$ with a target compression ratio $r$ is augmented with $C = \lceil L/r \rceil$ additional identical tokens. [1] The embedding of the compression token is learned. The final hidden state at the time steps of the compression tokens is taken as the compressed representation. A decoder, conditioned on this representation and a downstream prompt (e.g., a question), produces the output. Training typically combines a language modeling objective on the decoder with a distillation loss that encourages the compressor–decoder to approximate the behavior of a target LLM with access to the full uncompressed context (Figure 1a). The decoder parameters are either tuned during learning or are frozen.

This paradigm has been explored extensively in recent work. AutoCompressors (Chevalier et al., 2023) introduce recursive compression by appending a fixed set of compression tokens and extracting their hidden states. They tie the encoder and decoder weights. The ICAE framework (Ge et al., 2024) adopts an encoder–decoder setup where the decoder is frozen and only the encoder is trained, with a two-stage process of autoencoding pretraining and task-specific finetuning. COCOM (Rau et al., 2025) extends this approach to retrieval-augmented QA, experimenting with lighter encoders and with training decoders to jointly process multiple compressed contexts. Other work has sought more aggressive reduction: xRAG (Cheng et al., 2024a) maps document retrieval embeddings directly into the decoder's input space, achieving single-token compression but with severe constraints on sequence length and generality. PISCO (Louis et al., 2025) demonstrates that training compressors on LLM-generated answers improves downstream RAG performance, while PCC (Dai et al., 2025) decouples the compressor from the target LLM by learning a converter to project compressed representations into another model's hidden space. Most recently, GMSA (Tang et al., 2025) proposed grouping hidden representations and learning a layer semantic alignment module that bridges

---

[1]Although some works utilize a fixed number of compression tokens and then learn a distinct embedding for each position.

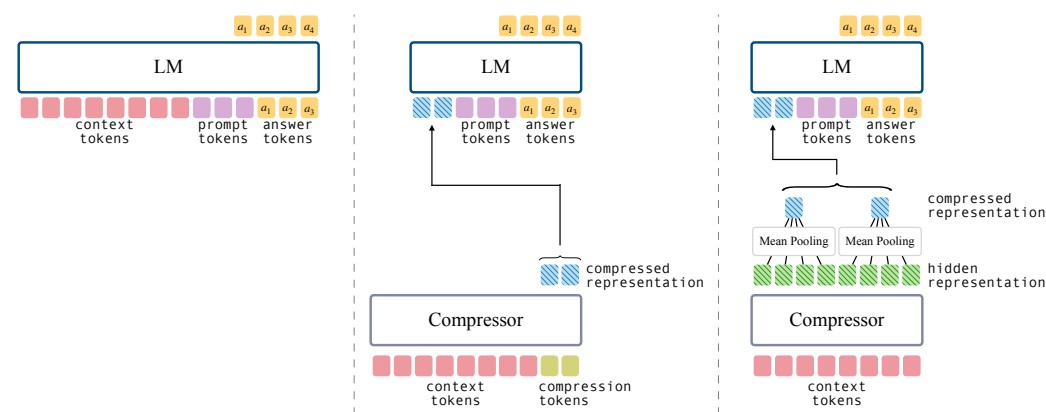

(a) Processing with a regular language model (no compression).

(b) Compression tokens approach for context compression.

(c) Our proposed method: no extra tokens; mean pooling of final hidden states.

Figure 1: Comparison of context processing methods: regular LM, compression tokens, and our proposed approach — mean pooling. The figure illustrates a compression ratio of $4\times$.

the gap between the encoder's final hidden states and the decoder's first attention layers. Their approach is related to our study of pooling, but differs in the complexity of multi-stage reconstruction training and the use of compression-decoder adapters.[2]

In this work, we revisit some of these design choices and provide a systematic comparison with token-based architectures under both single- and multi-ratio training regimes.

**KV Cache Compression**    In contrast to representing contexts as input embeddings, another line of work compresses the entire set of key–value (KV) states. Some approaches remove or compress less informative entries in the KV cache without additional training (Xiao et al., 2024; Oren et al., 2024; Li et al., 2024), while others train the model to perform the compression explicitly (Qin et al., 2024; Nawrot et al., 2024). A different variant introduces compression tokens, but instead of retaining only the final hidden representation, all KV states are propagated to the decoder (e.g., Zhang et al., 2025; Li et al., 2025). Although these methods provide higher-capacity compressed representations that are well suited for efficient long-context comprehension, their increased size makes it impractical to store them for reuse in retrieval-augmented generation frameworks, where caching compressed representations could otherwise avoid recomputation.

**Hard Prompt Compression**    An alternative approach is to compress contexts directly in the token space. This has been done by removing unimportant tokens or lexical units (e.g., Li et al., 2023; Jiang et al., 2023; Pan et al., 2024) or generating concise summaries that preserve salient details (Chuang et al., 2024). While these methods can be more interpretable and storage-efficient, they are inherently constrained by their reliance on explicit tokens.

## 4   METHODOLOGY

We propose a compression model design where representations of adjacent tokens are pooled together to reduce the document representation length. We train via knowledge distillation to replicate the functionality of a teacher receiving the original (uncompressed) input.

### 4.1   COMPRESSION VIA MEAN POOLING

We propose a simple compression architecture that relies only on mean pooling of encoded representations. Figure 1c illustrates our approach. Given a document, we compute its representation

---

[2]While we consider this approach an important point of comparison, we were unable to include it in our evaluation because their code and models were not publicly available at the time of writing.

with an encoder, and apply a non-overlapping mean pooling operator with window size $r$, the same size as the compression ratio, and stride $r$ to generate continuous vectors as the output compression.

Formally, let $h = (h_1, \ldots, h_L) \in \mathbb{R}^{L \times d}$ denote the sequence of hidden states produced by the encoder. For a compression ratio $r \in \mathcal{R}$, we partition the sequence into $k$ consecutive, non-overlapping blocks:

$$S_k = \{(k-1)r + 1, \ldots, \min(kr, L)\} \ , \qquad k = 1, \ldots, \lceil L/r \rceil \ . \tag{3}$$

The compressed representation of length $\lceil L/r \rceil$ is obtained by averaging within each block:

$$f_c(T, r) = (z_1, \ldots, z_{\lceil L/r \rceil}) \in \mathbb{R}^{\lceil L/r \rceil \times d} \quad \text{s.t.} \quad z_k = \frac{1}{|S_k|} \sum_{i \in S_k} h_i \ . \tag{4}$$

The encoder we use is Transformer-based. Critically, we use a full self-attention mask during encoding, allowing each encoded vector to include information from the entire context, and thereby each compressed segment to aggregate information across the entire context. In practice, we initialize with an autoregressive LLM, and remove the self-attention mask before learning.

Our pooling design introduces no additional parameters beyond those of the encoder backbone and the decoder (i.e., target LLM) using the compressed representation, and has low computational overhead. The compression-tokens method is slightly more expensive. It requires an encoder input of size $L + L/r$, while our approach only processes the original $L$ tokens, with negligible overhead for the pooling operator.

## 4.2 TRAINING OBJECTIVE

Our objective is to approximate the behavior of the target LLM before any fine-tuning when provided with the full uncompressed context (Figure 1a). We consider this LLM as the teacher model within a knowledge distillation framework. Each training instance consists of a context $T$, a prompt $P$, and a ground-truth answer $A = (a_1, \ldots, a_m)$. For each answer position $i = 1, \ldots, m$, the teacher defines the distribution $q(\cdot \mid T, P, A_{<i})$, where $A_{<i} = (a_1, \ldots, a_{i-1})$ denotes the gold prefix of the answer. The compressor produces a compressed representation $f_c(T, r)$, which is passed to the fine-tuned decoder together with $(P, A_{<i})$, yielding the student distribution $p_\theta(\cdot \mid f_c(T, r), P, A_{<i})$. This combined encoder-decoder model is the student.

For a single compression ratio $r$, the distillation loss is the token-level KL divergence between the teacher and student distributions:

$$\mathcal{L}_{\text{KD}}(T, P, A; r) = \sum_{i=1}^{m} \text{KL}(q(\cdot \mid T, P, A_{<i}) \parallel p_\theta(\cdot \mid f_c(T, r), P, A_{<i})) \ . \tag{5}$$

We propose a unified training strategy in which a single compressor is trained to handle multiple ratios simultaneously. This is in contrast to most previous work, where a separate model is trained for each compression ratio. We generate compressed representations for all ratios $r \in \mathcal{R}$ for each training instance. Each compressed representation is passed independently to the decoder, and the corresponding losses are computed. The final objective for one training instance is obtained by summing across the ratios:

$$\mathcal{L}_{\text{multi}}(T, P, A) = \sum_{r \in \mathcal{R}} \mathcal{L}_{\text{KD}}(T, P, A; r) \ . \tag{6}$$

The iteration over ratios is performed within each batch, and a single parameter update is applied after aggregating the losses. Since the encoder computation is shared across all ratios, this procedure is substantially more efficient than training separate models. By relying exclusively on knowledge distillation, rather than combining it with a language modeling objective, we enable a direct and fair comparison between the original model and the compressor.

## 5 EXPERIMENTS AND RESULTS

Our central objective is to isolate the contribution of context compression itself, without entanglement with retrieval noise or incomplete supervision. While compression methods are often demonstrated in retrieval-augmented generation (RAG) settings, these introduce extraneous challenges,

such as when retrieved passages may not contain sufficient evidence, making performance conflate retrieval quality with compression quality. To avoid this confounder, we focus exclusively on reading comprehension, where given contexts are guaranteed to contain the necessary evidence to answer the question. This setup allows for a controlled, head-to-head comparison of different compression strategies, across a range of datasets that stress both single-hop and multi-hop reasoning.

## 5.1 EXPERIMENTAL SETUP

**Data** We curated our training set by mixing multiple context-based datasets, in tasks spanning reading comprehension (RC) and summarization. A detailed list of the datasets we incorporated in our training mixture can be found in Table 4. We evaluate with six reading comprehension benchmarks: SQuAD (Rajpurkar et al., 2016), NarrativeQA (Kočiský et al., 2018), HotpotQA (Yang et al., 2018), AdversarialQA (Bartolo et al., 2020), TriviaQA (Joshi et al., 2017), and ParaphraseRC (Saha et al., 2018). This selection covers a broad spectrum of reasoning styles, from factual extraction to adversarial paraphrasing, thereby testing the generality of compression. For TriviaQA, we restrict the evaluation to the verified subset, ensuring that every question has sufficient supporting evidence. The training mixture includes the train splits of SQuAD, NarrativeQA, and HotpotQA, which thus serve as in-domain testbeds. AdversarialQA, TriviaQA, and ParaphraseRC are excluded from the training mixture and instead used purely for out-of-domain evaluation.

**Model Training** For each target language model, we first finetune it on the training mixture using LoRA (Hu et al., 2022). This finetuned model is then fixed and used as the teacher in the distillation process. This ensures that performance differences stem solely from the compressor rather than mismatched finetuning (e.g., to the question domain). Both the compressor's encoder and decoder are initialized from the same target LLM but are trained with separate LoRA weights. We always use the instruction-tuned model weights as our backbone. For multi-ratio training, we always train on the ratios $\{4\times, 8\times, 16\times, 32\times, 64\times, 128\times\}$, unless stated otherwise. In addition, we found that applying a single linear layer slightly improves performance for both our method and the compression-tokens method, so for all experiments in this paper a learned matrix $W \in \mathbb{R}^{d \times d}$ is applied to $f_c(T, r)$ before the compressed representation is given as input to the decoder LLM, unless stated otherwise. Detailed training configuration and hyperparameter choices are provided in Appendix A.

**Implementation of the Compression-Tokens Approach** The main approach we compare ours against is using compression tokens. A central design consideration in our experiments is the attention pattern applied to compression tokens. Under the compression-tokens paradigm, the causal attention mask typically employed by transformer-based LLMs imposes a strong Matryoshka-style (Kusupati et al., 2022) constraint: compressed representations at smaller lengths must correspond to strict prefixes of those at larger lengths. We relax this restriction by allowing compression tokens to attend bidirectionally among themselves, while retaining causal attention over the original context. This simple, albeit not explored in prior work, modification makes the model aware of how many compression tokens are available (i.e., its compression budget), and therefore to allocate information differently for each ratio while still benefiting from shared computation and KV caching. We experiment with both the conventional causal attention mask and our bidirectional modification. Empirically, we observe that this modification significantly improves the approach's performance. In addition, in our implementation of the compression-tokens models, we utilize only a single compression token that is appended $\lceil L/r \rceil$ times to the context, rather than having several compression tokens. This enables us to compress any context to any arbitrary ratio we choose, while retaining an equal number of parameters in the model.

**Metrics** We evaluate our models using the standard *exact match* (EM) and $F_1$ metrics.[3] Several recent works reported QA performance using a substring accuracy metric, which assigns a score of 1 if the exact match is a substring of the output and 0 otherwise. We opt against the adoption of this metric as it is easily exploitable.[4] This forces us to exclude some baselines from our primary results.

In addition, for each metric, we also define its *teacher-normalized* version. Given a metric $M$, a target language model $\mathcal{M}$ and a compressor $f_c$, we calculate the following: $M_T$, the teacher's score,

---

[3] We show only $F_1$ scores in the main text but full EM results with similar trends are reported in Table 6.

[4] Consider a question where the answer is a US state, and the model lists all 50 states as an answer.

by passing the original uncompressed contexts to the decoder model; $M_T^\varnothing$, the no-context score, by passing only the question without any context; and $M_{f_c}$, by using the compressed context. Then, the teacher-normalized score is given by $\frac{M_{f_c} - M_T^\varnothing}{M_T - M_T^\varnothing}$. This score does two things. First, it scales the compressor's score with respect to the teacher's, allowing for a direct assessment of the amount of retained performance under compression. Second, it accounts for how easy it is for the model to answer the input question without any context. This latter consideration comes to account for cases where a question does not really require the context, potentially inflating the score of the compression model, while actually simply ignoring the compressed input.

**Systems** Our main comparison point is our implementations of the compression tokens method, with causal or bidirectional attention. We compare against these two systems throughout our experiments. We also include comparisons to other soft context compression methods: ICAE (Ge et al., 2024) and PCC (Dai et al., 2025). We also evaluate LLMLingua2 (Pan et al., 2024), a hard prompting compression approach, by passing LLMLingua's compressed prompts to our finetuned Qwen3-8B teacher model.

Comparison between compression methods is challenging in general, due to inconsistencies in the training procedures. Our main goal is to build a systematic understanding of the architecture landscape, and we compare ourselves to other methods mainly to showcase the validity of our experiments. Showing the strength of performance compared to the prior state of the art is secondary, and not even necessarily feasible because many approaches do not release code or models, or adopt training methods that complicate the evaluation. This challenge is demonstrated well by the PISCO (Louis et al., 2025) method. Their training method is not well suited for the traditional EM/$F_1$ metrics and performs poorly on these metrics, and indeed the authors evaluated their method only using a more forgiving substring accuracy score. We therefore omit this method from our primary comparisons, but include it and report substring accuracy in Appendix B.1 (Table 7).

All experiments are conducted with context lengths up to 1,024 tokens, a practical limitation imposed by our computational budget.

## 5.2 RESULTS

Table 1 shows our main results. We demonstrate the robustness and generality of our findings by comparing six different models, spanning three model families and four model scales: Llama3.2-1B (Grattafiori et al., 2024), Gemma2-2b (Team et al., 2024), and Qwen3-0.6/1.7/4/8B (Yang et al., 2025). We show the average $F_1$ scores over all six evaluation datasets. Below the results table, we provide bar charts that summarize the results along specific dimensions that emphasize trends. The bar charts show the teacher-normalized $F_1$ metric averaged across all models and datasets to get a single aggregated result per method. We provide the teacher model's performance with the original context ("Original") and without context at all ("No Ctx"). In between we report the results for different compression ratios and for both the single- and multi-ratio training schemes.

Our mean-pooling method is consistently better than both the standard (causal) compression tokens architecture and the bidirectional variation. Simply adding bidirectional attention between the compression tokens dramatically improves performance. Comparing the single-ratio models to the multi-ratio models, we see that the bidirectional compression tokens approach significantly benefits from multi-ratio training, while for the mean-pooling approach a trade-off clearly exists. This is potentially because the bidirectional approach has a clear signal about the target compression ratio – the forward attention allows each time step to be aware of the total length of the compressed output. Without this, the model must produce a one-size-fits-all representation in each time step. Lastly, while we do see a performance drop in the multi-ratio setting at $128\times$ for all methods, it must be taken in the context of the already relatively low performance retention at that ratio.

**Compression Scaling** It is well known that LLM performance increases with scale, if scaled appropriately (Cheng et al., 2024b). But does compression quality scale as well? In Table 1 we can see that the teacher's performance improves as the model size grows. Having a compressor increase in performance at the same rate as the teacher would actually tell us that the compressor does not scale well, since that would mean that the teacher-normalized scores stay constant throughout the scales. Figure 2 shows the compression performances of the four Qwen3 model scales we use. We train

| | Original | 4x | | 16x | | 128x | | No Ctx |
|---|---|---|---|---|---|---|---|---|
| | | **Single** | **Multi** | **Single** | **Multi** | **Single** | **Multi** | |
| **Baseline Systems** | | | | | | | | |
| *LLMLingua2* (Qwen3-8B) | | | 42.52 | | 24.39 | | 22.59 | |
| *ICAE* (Mistral-7B) | | 42.40 | | | | | | |
| *PCC Lite* (GPT2-Large & Llama3.1-8B) | | 62.08 | | 51.30 | | 36.20 | | |
| *PCC Large* (Llama3.1-8B) | | 62.98 | | 49.37 | | 37.24 | | |
| **Our Methods** | | | | | | | | |
| **Qwen3-8B** | 74.33 | | | | | | | 23.06 |
| Compression-Tokens (Causal) | | 67.03 | 65.90 | 56.21 | 58.41 | 47.47 | 44.76 | |
| Compression-Tokens (Bidirectional) | | 69.20 | 69.57 | 60.27 | 63.01 | 46.93 | **46.97** | |
| Mean-Pooling | | **71.66** | **70.55** | **63.85** | **64.67** | **47.90** | 45.92 | |
| **Qwen3-4B** | 73.44 | | | | | | | 19.79 |
| Compression-Tokens (Causal) | | 64.88 | 62.53 | 55.22 | 54.28 | 43.08 | 40.83 | |
| Compression-Tokens (Bidirectional) | | 66.72 | 68.15 | 57.68 | 60.48 | 41.61 | **42.66** | |
| Mean-Pooling | | **70.39** | **69.36** | **61.79** | **61.72** | **43.62** | 41.05 | |
| **Qwen3-1.7B** | 69.93 | | | | | | | 14.00 |
| Compression-Tokens (Causal) | | 50.90 | 57.73 | 49.83 | 48.68 | 36.19 | 35.34 | |
| Compression-Tokens (Bidirectional) | | 62.04 | 62.60 | 51.53 | 54.11 | 36.25 | **35.77** | |
| Mean-Pooling | | **66.43** | **64.17** | **55.43** | **54.47** | **36.72** | 33.48 | |
| **Qwen3-0.6B** | 65.36 | | | | | | | 9.34 |
| Compression-Tokens (Causal) | | 54.40 | 51.85 | 41.57 | 42.59 | 28.86 | 28.60 | |
| Compression-Tokens (Bidirectional) | | 55.59 | 57.03 | 44.82 | 47.62 | 29.69 | **29.51** | |
| Mean-Pooling | | **61.17** | **58.36** | **47.59** | **47.64** | **29.94** | 26.36 | |
| **Gemma2-2B** | 71.96 | | | | | | | 21.64 |
| Compression-Tokens (Causal) | | 63.35 | 62.18 | 55.07 | 54.70 | 44.46 | 42.49 | |
| Compression-Tokens (Bidirectional) | | 64.76 | 65.24 | 56.39 | 58.43 | 44.73 | 43.17 | |
| Mean-Pooling | | **69.33** | **68.09** | **61.39** | **61.04** | **44.98** | **43.71** | |
| **Llama3.2-1B** | 65.82 | | | | | | | 15.17 |
| Compression-Tokens (Causal) | | 56.31 | 53.74 | 47.51 | 46.96 | 35.41 | 35.62 | |
| Compression-Tokens (Bidirectional) | | 57.91 | 57.52 | **49.20** | 50.06 | **36.43** | **36.25** | |
| Mean-Pooling | | **62.81** | **60.56** | 47.28 | **51.56** | 33.25 | 33.98 | |

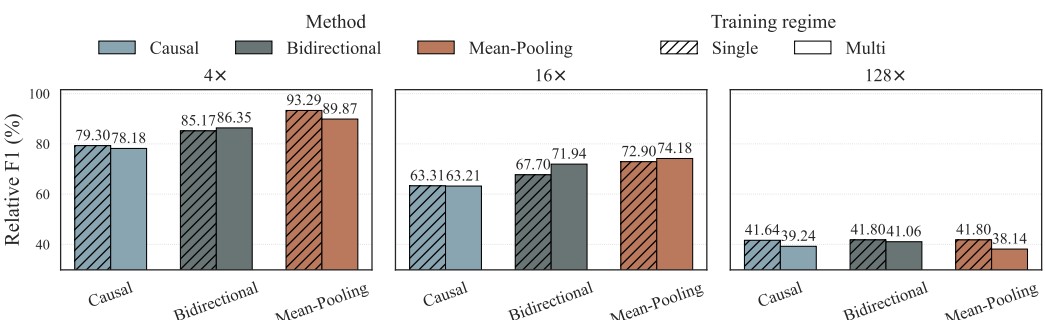

Table 1: Primary results. Values in the table are $F_1$ scores macro-averaged across all datasets in our evaluation suite. **Original** stands for the teacher model's score when given the full context. **No Ctx** stands for the teacher model's score when not given any context at all. For each ratio, we display both single- and multi-ratio versions. For the baseline systems (top section), we include results for the compression ratios supported by these methods, unsupported ratios are left blank. The best method for each (model, ratio, single/multi-ratio training) setting is **bolded**. Bottom figures present aggregated views of the results in the table, but instead of $F_1$ show the teacher-normalized $F_1$ metric (Relative F1). Scores are obtained by averaging over all models listed in the table.

the models under the multi-ratio setting to efficiently evaluate multiple ratios. We evaluate using the teacher-normalized $F_1$ score, and present the average scores across all datasets. The results are exciting—the compressors clearly have desirable scaling properties. Critically, the efficiency gains of context compression are much larger as the model size increases. Therefore, observing that larger

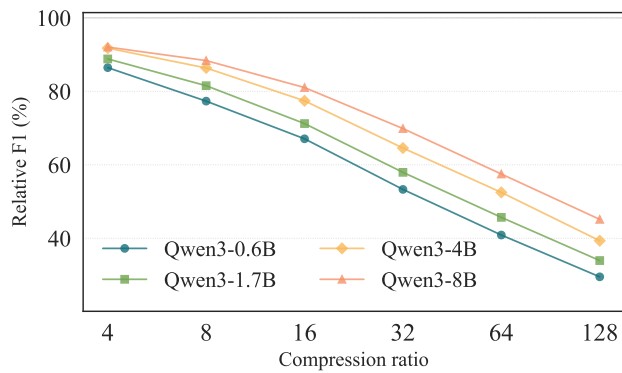

Figure 2: Compression Scaling. We show the teacher-normalized $F_1$ scores (Relative F1) across four Qwen3 model scales. The scores are averages of the scores of all datasets. We can clearly observe the benefits of scaling for LLM compressors.

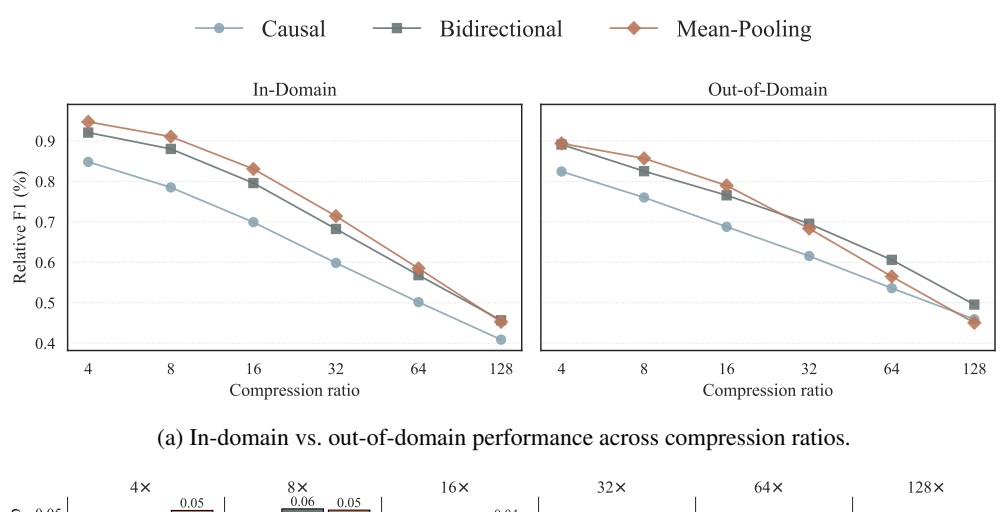

(a) In-domain vs. out-of-domain performance across compression ratios.

(b) Performance drop (in–out gap) per method across compression ratios (in teacher-normalized Relative $F_1$ units). Higher values mean a larger domain performance gap. Negative values mean that the out-of-domain performance is better than in-domain performance.

Figure 3: In-domain and out-of-domain comparison. (a) Line plots show performance on in-domain vs. out-of-domain datasets. (b) Bar plots show the in–out performance gap per method.

models retain higher levels of performance (higher relative F1) justifies this line of research. Similar trends are observed for the other methods (Table 1), demonstrating the generality of this finding.

**In-Domain vs. Out-of-Domain** We construct our evaluation suite with both in-domain QA datasets and out-of-domain QA datasets (Section 5.1). The training splits of the in-domain datasets are included in the training data mixture, while the out-of-domains datasets are not. It is expected that downstream performance will drop for out-of-domain datasets. Critical for our study, though, is the performance drop of the compressor itself. Figure 3a plots the in-domain and out-of-domain performance using the teacher-normalized $F_1$ score for the Qwen3-8B model, averaged over the datasets in each category. We first observe that while the mean-pooling approach is superior for ratios up to $16\times$ in all settings, its performance deteriorates as the compression ratio increases. To

| Ablation (GEMMA2-2B) | 4× | 8× | 16× | 32× | 64× | 128× | Δ |
|---|---|---|---|---|---|---|---|
| Default | **68.1** | **65.4** | **61.0** | **54.9** | **48.8** | **43.7** | (+0.0) |
| Fixed Decoder | 64.9 | 61.9 | 57.0 | 51.5 | 45.0 | 39.8 | (−3.6) |
| Fixed Encoder | 57.4 | 49.9 | 44.1 | 39.8 | 36.2 | 34.8 | (−13.3) |
| No Encoder | 58.7 | 51.9 | 44.9 | 40.2 | 36.2 | 34.1 | (−12.6) |
| w/o Linear Layer | 67.7 | 64.5 | 60.0 | 54.1 | 48.1 | 43.2 | (−0.7) |
| Ratio Sampling | 67.1 | 64.0 | 59.3 | 53.5 | 47.5 | 42.2 | (−1.4) |

Table 2: Ablation study for mean pooling using GEMMA2-2B as the teacher LLM. Numbers are macro-averaged $F_1$ scores. Δ: mean change vs. Default across ratios; **bold** = best per column.

better understand the performance change due to the domain gap, we plot the differences between the in-domain and out-of-domain performance in Figure 3b. The performance gap is higher for low ratios, and lower for higher ratios. One possible explanation is that at low compression ratios the compressed representations still retain much of the original contextual signal, so the model is more sensitive to domain-specific distributional shifts; differences between in-domain and out-of-domain language patterns thus manifest as a larger performance gap. By contrast, at higher compression ratios much of the fine-grained contextual detail is already lost to compression noise, which dominates over the domain gap. In this regime, both in-domain and out-of-domain datasets suffer similarly from the limited representational capacity, resulting in a smaller relative difference.

**Model Ablations**  We run all our ablations using the Gemma2-2B model, since it demonstrates strong performance while being relatively compute-efficient. Table 2 presents the ablation results. We conduct several ablations: (1) Fixed Decoder: the decoder is kept frozen and only the encoder is trained; (2) Fixed Encoder: the encoder is kept frozen and only the decoder is trained; (3) No Encoder: we remove the encoder entirely, and obtain the initial context representation using only the token embeddings of the decoder model; (4) w/o Linear Layer: we remove the linear layer that is applied after the pooling operation; (5) Ratio Sampling: instead of training on all ratios for each sample, a single ratio is randomly chosen for each sample during training.

Freezing the decoder results in considerable performance reduction, although not catastrophic. This is in line with findings of previous works ((Louis et al., 2025)). Freezing or removing the encoder is more detrimental, lowering performance by more than 12%. The effect of the linear layer is not very significant, its removal results in a reduction of only 0.7%. Finally, while randomly sampling a single ratio per sample would speed up training significantly, it does so at the cost of a small drop in performance (1.4%), likely because each ratio gets less updates during training.

## 6  DISCUSSION

We provide a systematic study of soft context compression, showing that a simple mean-pooling approach consistently outperforms compression-tokens architectures while requiring no additional parameters. We further demonstrate that multi-ratio training is both feasible and effective, enabling a single compressor to support a wide range of compression budgets with only minor performance degradation. Interestingly, we observe that the bidirectional compression-tokens method consistently benefits from multi-ratio training. A plausible explanation is that, unlike mean-pooling or causal compression-tokens, this method has explicit access to the number of compression tokens available, allowing it to adapt to the target budget. Exploring how to incorporate similar signals into other architectures is an important direction for future work.

Finally, our study highlights a broader open problem: the evaluation of compression methods remains hindered by inconsistent setups, metrics, and benchmarks. By isolating compression quality from retrieval confounders and applying a uniform evaluation across models, scales, and domains, we aim to provide a rigorous basis for future research on soft context compression. We hope this work helps establish more standardized practices and clarifies the core principles that should guide the development of next-generation compressors.

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

# A ADDITIONAL EXPERIMENTAL SETUP

## A.1 TRAINING HYPERPARAMETERS

| Hyperparameter | Value |
|---|---|
| LoRA $r$ | 16 |
| LoRA $\alpha$ | 16 |
| optimizer | AdamW |
| $\beta_1$ | 0.9 |
| $\beta_2$ | 0.95 |
| clip norm | 1 |
| peak learning rate | 2e-4 |
| final learning rate | 2e-5 |
| lr scheduler type | cosine |
| warmup ratio | 0.05 |
| weight decay | 0.0 |
| steps | 48,000 |
| batch size | 32 |
| max context length | 1024 |
| max answer tokens | 256 |

Table 3: Hyperparameters for training all the models presented in this paper.

## A.2 DATA

Detailed lists of our training data mixture and evaluation suite are given in Table 4 and Table 5, respectively. For NarrativeQA, we use the summaries as contexts, instead of the full stories. For HotpotQA, we only use the two gold paragraphs as contexts, and remove the distractors.

When training on a sample consisting of a context $C$, question $Q$ and answer $A$, we randomly sample a prompt template that fits the task to make the training data more diverse. For example, for the extractive QA task, an example of a prompt template is: *"$<C>\backslash n$ Extract the answer from the text above. $\backslash n$ Question: $<Q>\backslash n$ Answer: $<A>$"*. Similar templates are defined for other tasks as well. For each task, we created approximately 100 prompt templates. The full list of templates for each task will be released along with our code.

| Dataset | Avg. Context Tokens | #Samples | #Contexts |
|---|---|---|---|
| **Summarization** | | | |
| CNN/DM (See et al., 2017) | 649 | 198,732 | 196,601 |
| DialogSum (Chen et al., 2021) | 208 | 12,452 | 12,450 |
| SAMSum (Gliwa et al., 2019) | 145 | 14,730 | 14,254 |
| XSum (Narayan et al., 2018) | 408 | 185,760 | 185,566 |
| **Reading Comprehension** | | | |
| BoolQ (Clark et al., 2019) | 126 | 9,427 | 7,927 |
| DROP (Dua et al., 2019) | 295 | 76,751 | 5,477 |
| HotpotQA (Yang et al., 2018) | 247 | 90,327 | 84,705 |
| NarrativeQA (Kočiský et al., 2018) | 668 | 28,299 | 953 |
| PubMedQA (Jin et al., 2019) | 318 | 211,218 | 211,164 |
| QuAC (Choi et al., 2018) | 515 | 81,391 | 6,574 |
| QuAIL (Rogers et al., 2020) | 416 | 10,246 | 560 |
| RACE (Lai et al., 2017) | 349 | 87,749 | 25,108 |
| SQuAD (Rajpurkar et al., 2016) | 162 | 86,821 | 18,877 |
| PWC (Ge et al., 2024) | 477 | 241,563 | 16,382 |
| **Total** | **410** | **1,335,466** | **786,598** |

Table 4: Training datasets with average context length (tokens), number of samples, number of distinct contexts, and task category. The overall average context length is weighted by number of samples.

| Dataset | Avg. Context Tokens | #Samples | #Contexts |
|---|---|---|---|
| AdversarialQA (Bartolo et al., 2020) | 154 | 1,000 | 341 |
| HotpotQA (Yang et al., 2018) | 254 | 7,394 | 7,352 |
| NarrativeQA (Kočiský et al., 2018) | 639 | 3,002 | 100 |
| ParaphraseRC (Saha et al., 2018) | 685 | 4,835 | 560 |
| SQuAD (Rajpurkar et al., 2016) | 169 | 5,928 | 1,204 |
| TriviaQA (Verified) (Joshi et al., 2017) | 539 | 185 | 185 |
| **Total** | **375** | **22,344** | **9,742** |

Table 5: Evaluation datasets with average context length (tokens), number of samples, and number of distinct contexts. The overall average context length is weighted by number of samples.

# B  ADDITIONAL RESULTS

In this section we provide additional results from the same experiments conducted in the main body of the paper. In Appendix B.1 we provide the primary results of the paper with the EM and substring accuracy metrics. In Appendix B.2 we show the $F_1$ performance on each individual dataset from the evaluation suite.

## B.1 PRIMARY RESULTS —ADDITIONAL METRICS

| | Original | 4x | | 16x | | 128x | | No Ctx |
|---|---|---|---|---|---|---|---|---|
| | | **Single** | **Multi** | **Single** | **Multi** | **Single** | **Multi** | |
| **Baseline Systems** | | | | | | | | |
| *LLMLingua2* (Qwen3-8B) | | | 30.02 | | 16.16 | | 16.06 | |
| *ICAE* (Mistral-7B) | | 24.94 | | | | | | |
| *PCC Lite* (GPT2-Large & Llama3.1-8B) | | 48.81 | | 38.38 | | 25.94 | | |
| *PCC Large* (Llama3.1-8B) | | 49.34 | | 36.64 | | 27.92 | | |
| **Our Methods** | | | | | | | | |
| **Qwen3-8B** | 59.82 | | | | | | | 16.19 |
| Compression-Tokens (Causal) | | 51.59 | 50.01 | 41.26 | 42.99 | 34.21 | 32.50 | |
| Compression-Tokens (Bidirectional) | | 53.44 | 53.99 | 44.92 | 47.15 | 33.60 | 34.27 | |
| Mean-Pooling | | 56.41 | 55.04 | 47.95 | 49.02 | 34.72 | 33.15 | |
| **Qwen3-4B** | 58.87 | | | | | | | 13.74 |
| Compression-Tokens (Causal) | | 49.66 | 46.26 | 40.05 | 39.59 | 30.07 | 28.84 | |
| Compression-Tokens (Bidirectional) | | 51.06 | 52.37 | 42.53 | 45.14 | 28.84 | 29.51 | |
| Mean-Pooling | | 55.25 | 53.77 | 45.43 | 45.86 | 30.91 | 28.38 | |
| **Qwen3-1.7B** | 55.19 | | | | | | | 9.07 |
| Compression-Tokens (Causal) | | 36.66 | 41.64 | 35.49 | 34.64 | 24.27 | 24.04 | |
| Compression-Tokens (Bidirectional) | | 46.45 | 46.72 | 36.62 | 38.93 | 24.31 | 24.33 | |
| Mean-Pooling | | 51.28 | 48.77 | 40.45 | 39.04 | 25.15 | 22.01 | |
| **Qwen3-0.6B** | 50.85 | | | | | | | 4.78 |
| Compression-Tokens (Causal) | | 39.66 | 36.76 | 27.99 | 29.00 | 18.23 | 18.20 | |
| Compression-Tokens (Bidirectional) | | 40.98 | 41.92 | 30.92 | 33.64 | 18.82 | 18.55 | |
| Mean-Pooling | | 45.82 | 43.07 | 32.50 | 33.09 | 19.05 | 16.07 | |
| **Gemma2-2B** | 57.63 | | | | | | | 15.00 |
| Compression-Tokens (Causal) | | 47.90 | 45.98 | 39.57 | 39.74 | 32.02 | 29.66 | |
| Compression-Tokens (Bidirectional) | | 49.43 | 49.40 | 40.89 | 42.70 | 31.79 | 30.43 | |
| Mean-Pooling | | 54.20 | 52.77 | 45.88 | 45.68 | 32.41 | 30.83 | |
| **Llama3.2-1B** | 51.67 | | | | | | | 9.47 |
| Compression-Tokens (Causal) | | 41.84 | 39.03 | 33.73 | 33.38 | 24.11 | 24.64 | |
| Compression-Tokens (Bidirectional) | | 43.46 | 42.96 | 35.04 | 35.46 | 24.91 | 24.56 | |
| Mean-Pooling | | 47.97 | 45.45 | 33.15 | 36.93 | 22.89 | 22.70 | |

Table 6: Primary results with exact match (EM) as the metric.

| | Original | 4x | | 16x | | 128x | | No Ctx |
|---|---|---|---|---|---|---|---|---|
| | | Single | Multi | Single | Multi | Single | Multi | |
| **Baseline Systems** | | | | | | | | |
| *LLMLingua2* (Qwen3-8B) | | | 33.63 | | 18.30 | | 17.36 | |
| *ICAE* (Mistral-7B) | | 49.18 | | | | | | |
| *PISCO* (Llama3.1-8B) | | | | 53.62 | | | | |
| *PCC Lite* (GPT2-Large & Llama3.1-8B) | | 54.05 | | 43.67 | | 30.03 | | |
| *PCC Large* (Llama3.1-8B) | | 55.17 | | 41.79 | | 30.10 | | |
| **Our Methods** | | | | | | | | |
| **Qwen3-8B** | 68.84 | | | | | | | 17.98 |
| Compression-Tokens (Causal) | | 59.79 | 58.58 | 47.07 | 49.99 | 39.09 | 37.05 | |
| Compression-Tokens (Bidirectional) | | 62.50 | 63.58 | 52.22 | 55.26 | 38.72 | 39.36 | |
| Mean-Pooling | | 65.91 | 65.06 | 55.68 | 56.77 | 39.95 | 37.80 | |
| **Qwen3-4B** | 67.69 | | | | | | | 15.00 |
| Compression-Tokens (Causal) | | 57.12 | 54.17 | 46.06 | 45.47 | 34.83 | 33.48 | |
| Compression-Tokens (Bidirectional) | | 59.35 | 60.99 | 48.92 | 52.23 | 33.20 | 34.33 | |
| Mean-Pooling | | 64.05 | 62.94 | 52.77 | 52.82 | 35.49 | 32.37 | |
| **Qwen3-1.7B** | 64.21 | | | | | | | 9.85 |
| Compression-Tokens (Causal) | | 43.01 | 49.31 | 41.31 | 40.45 | 28.45 | 27.90 | |
| Compression-Tokens (Bidirectional) | | 54.31 | 54.70 | 42.66 | 45.13 | 28.33 | 28.04 | |
| Mean-Pooling | | 60.03 | 57.28 | 46.63 | 45.07 | 28.96 | 25.82 | |
| **Qwen3-0.6B** | 59.67 | | | | | | | 5.62 |
| Compression-Tokens (Causal) | | 46.36 | 43.62 | 33.07 | 34.30 | 21.13 | 21.71 | |
| Compression-Tokens (Bidirectional) | | 48.04 | 48.84 | 36.11 | 39.50 | 22.13 | 22.00 | |
| Mean-Pooling | | 54.36 | 51.16 | 38.30 | 38.68 | 22.44 | 18.87 | |
| **Gemma2-2B** | 66.14 | | | | | | | 16.80 |
| Compression-Tokens (Causal) | | 55.50 | 52.94 | 46.13 | 45.59 | 36.14 | 33.94 | |
| Compression-Tokens (Bidirectional) | | 56.94 | 57.54 | 46.90 | 49.52 | 36.14 | 34.75 | |
| Mean-Pooling | | 62.51 | 61.28 | 52.55 | 51.90 | 36.61 | 34.93 | |
| **Llama3.2-1B** | 60.30 | | | | | | | 11.09 |
| Compression-Tokens (Causal) | | 48.85 | 45.41 | 39.16 | 38.51 | 27.54 | 28.01 | |
| Compression-Tokens (Bidirectional) | | 50.89 | 50.10 | 40.44 | 41.91 | 28.34 | 28.44 | |
| Mean-Pooling | | 56.62 | 53.50 | 38.84 | 42.76 | 25.99 | 26.19 | |

Table 7: Primary results with accuracy as the metric.

## B.2   RESULTS PER DATASET

### B.2.1   IN DOMAIN DATASETS RESULTS

| | Original | 4x | | 16x | | 128x | | No Ctx |
|---|---|---|---|---|---|---|---|---|
| | | Single | Multi | Single | Multi | Single | Multi | |
| **Baseline Systems** | | | | | | | | |
| *LLMLingua2* (Qwen3-8B) | | | 48.38 | | 21.34 | | 19.68 | |
| *ICAE* (Mistral-7B) | | 45.6 | | | | | | |
| *PCC Lite* (GPT2-Large & Llama3.1-8B) | | 78.38 | | 67.63 | | 40.22 | | |
| *PCC Large* (Llama3.1-8B) | | 79.56 | | 62.93 | | 41.23 | | |
| **Our Methods** | | | | | | | | |
| **Qwen3-8B** | 86.48 | | | | | | | 20.31 |
| Compression-Tokens (Causal) | | 77.11 | 74.89 | 57.05 | 62.12 | 44.56 | 42.35 | |
| Compression-Tokens (Bidirectional) | | 80.00 | 81.23 | 64.80 | 69.27 | 44.30 | 43.86 | |
| Mean-Pooling | | 83.76 | 82.75 | 71.37 | 71.19 | 44.65 | 43.19 | |
| **Qwen3-4B** | 85.75 | | | | | | | 17.72 |
| Compression-Tokens (Causal) | | 74.23 | 71.31 | 57.49 | 56.88 | 38.95 | 37.10 | |
| Compression-Tokens (Bidirectional) | | 77.24 | 79.28 | 60.71 | 65.49 | 38.19 | 38.41 | |
| Mean-Pooling | | 83.19 | 81.54 | 68.20 | 67.47 | 39.96 | 37.54 | |
| **Qwen3-1.7B** | 83.65 | | | | | | | 12.66 |
| Compression-Tokens (Causal) | | 54.25 | 64.50 | 49.09 | 49.98 | 31.78 | 30.10 | |
| Compression-Tokens (Bidirectional) | | 72.92 | 73.39 | 53.07 | 57.36 | 31.58 | 31.17 | |
| Mean-Pooling | | 79.56 | 77.17 | 59.01 | 58.30 | 32.36 | 29.90 | |
| **Qwen3-0.6B** | 81.55 | | | | | | | 7.91 |
| Compression-Tokens (Causal) | | 61.67 | 57.52 | 40.29 | 41.60 | 22.06 | 22.45 | |
| Compression-Tokens (Bidirectional) | | 64.21 | 67.05 | 42.81 | 46.68 | 22.09 | 22.36 | |
| Mean-Pooling | | 74.00 | 70.60 | 48.62 | 48.14 | 22.40 | 20.45 | |
| **Gemma2-2B** | 84.58 | | | | | | | 16.41 |
| Compression-Tokens (Causal) | | 70.61 | 69.06 | 55.75 | 56.37 | 37.89 | 35.66 | |
| Compression-Tokens (Bidirectional) | | 74.16 | 75.41 | 57.77 | 61.75 | 37.72 | 37.00 | |
| Mean-Pooling | | 81.67 | 80.01 | 66.38 | 65.64 | 38.96 | 36.71 | |
| **Llama3.2-1B** | 81.16 | | | | | | | 11.27 |
| Compression-Tokens (Causal) | | 62.65 | 59.34 | 46.41 | 45.49 | 28.58 | 28.44 | |
| Compression-Tokens (Bidirectional) | | 64.48 | 65.61 | 48.99 | 51.39 | 29.09 | 28.82 | |
| Mean-Pooling | | 74.91 | 71.40 | 47.36 | 53.66 | 28.02 | 27.91 | |

Table 8: SQuAD $F_1$

| | Original | 4x | | 16x | | 128x | | No Ctx |
|---|---|---|---|---|---|---|---|---|
| | | Single | Multi | Single | Multi | Single | Multi | |
| **Baseline Systems** | | | | | | | | |
| *LLMLingua2* (Qwen3-8B) | | | 53.54 | | 29.32 | | 26.27 | |
| *ICAE* (Mistral-7B) | | 50.01 | | | | | | |
| *PCC Lite* (GPT2-Large & Llama3.1-8B) | | 68.55 | | 59.38 | | 43.93 | | |
| *PCC Large* (Llama3.1-8B) | | 70.08 | | 59.05 | | 46.46 | | |
| **Our Methods** | | | | | | | | |
| **Qwen3-8B** | 84.67 | | | | | | | 26.65 |
| Compression-Tokens (Causal) | | 78.85 | 78.32 | 68.00 | 72.65 | 64.14 | 59.74 | |
| Compression-Tokens (Bidirectional) | | 80.24 | 81.45 | 73.30 | 76.77 | 63.26 | 62.44 | |
| Mean-Pooling | | 83.30 | 82.08 | 77.66 | 78.41 | 63.88 | 63.77 | |
| **Qwen3-4B** | 84.12 | | | | | | | 23.16 |
| Compression-Tokens (Causal) | | 76.48 | 75.56 | 68.85 | 69.80 | 59.08 | 55.78 | |
| Compression-Tokens (Bidirectional) | | 78.13 | 79.41 | 71.11 | 74.60 | 58.38 | 56.87 | |
| Mean-Pooling | | 82.20 | 80.77 | 75.45 | 76.02 | 59.29 | 58.74 | |
| **Qwen3-1.7B** | 80.95 | | | | | | | 18.75 |
| Compression-Tokens (Causal) | | 66.69 | 70.98 | 63.82 | 63.84 | 51.48 | 48.78 | |
| Compression-Tokens (Bidirectional) | | 73.73 | 74.93 | 66.06 | 68.50 | 52.08 | 50.77 | |
| Mean-Pooling | | 78.64 | 76.11 | 68.76 | 68.78 | 50.86 | 49.44 | |
| **Qwen3-0.6B** | 77.35 | | | | | | | 14.74 |
| Compression-Tokens (Causal) | | 66.62 | 65.76 | 55.88 | 57.79 | 43.16 | 39.58 | |
| Compression-Tokens (Bidirectional) | | 67.24 | 69.28 | 58.44 | 61.79 | 43.80 | 41.66 | |
| Mean-Pooling | | 73.00 | 69.58 | 61.13 | 61.08 | 42.76 | 40.45 | |
| **Gemma2-2B** | 82.55 | | | | | | | 25.18 |
| Compression-Tokens (Causal) | | 75.62 | 75.54 | 69.18 | 70.42 | 61.91 | 59.03 | |
| Compression-Tokens (Bidirectional) | | 76.55 | 77.60 | 69.64 | 73.51 | 61.67 | 60.46 | |
| Mean-Pooling | | 80.93 | 79.85 | 75.10 | 74.90 | 62.36 | 61.64 | |
| **Llama3.2-1B** | 77.96 | | | | | | | 19.34 |
| Compression-Tokens (Causal) | | 69.68 | 68.22 | 63.36 | 62.96 | 52.27 | 49.81 | |
| Compression-Tokens (Bidirectional) | | 70.74 | 71.01 | 64.47 | 66.19 | 53.51 | 51.79 | |
| Mean-Pooling | | 74.38 | 72.69 | 62.75 | 66.59 | 51.05 | 50.86 | |

Table 9: HotpotQA $F_1$

| | Original | 4x | | 16x | | 128x | | No Ctx |
|---|---|---|---|---|---|---|---|---|
| | | Single | Multi | Single | Multi | Single | Multi | |
| **Baseline Systems** | | | | | | | | |
| *LLMLingua2* (Qwen3-8B) | | | 27.33 | | 14.35 | | 10.69 | |
| *ICAE* (Mistral-7B) | | 32.65 | | | | | | |
| *PCC Lite* (GPT2-Large & Llama3.1-8B) | | 50.29 | | 34.16 | | 16.05 | | |
| *PCC Large* (Llama3.1-8B) | | 50.72 | | 32.56 | | 16.18 | | |
| **Our Methods** | | | | | | | | |
| **Qwen3-8B** | 68.00 | | | | | | | 10.93 |
| Compression-Tokens (Causal) | | 59.62 | 58.28 | 46.58 | 49.32 | 33.41 | 29.37 | |
| Compression-Tokens (Bidirectional) | | 61.44 | 62.13 | 51.48 | 55.68 | 34.17 | 33.61 | |
| Mean-Pooling | | 65.89 | 64.74 | 58.17 | 58.38 | 34.81 | 32.21 | |
| **Qwen3-4B** | 67.12 | | | | | | | 10.40 |
| Compression-Tokens (Causal) | | 57.18 | 55.08 | 46.69 | 43.77 | 30.39 | 27.84 | |
| Compression-Tokens (Bidirectional) | | 58.37 | 60.74 | 48.84 | 52.41 | 29.77 | 30.22 | |
| Mean-Pooling | | 65.22 | 63.38 | 56.14 | 55.42 | 32.66 | 28.99 | |
| **Qwen3-1.7B** | 64.42 | | | | | | | 7.57 |
| Compression-Tokens (Causal) | | 41.97 | 49.85 | 40.15 | 39.49 | 25.42 | 23.64 | |
| Compression-Tokens (Bidirectional) | | 55.68 | 56.49 | 44.87 | 47.28 | 25.16 | 26.15 | |
| Mean-Pooling | | 60.34 | 58.56 | 48.95 | 48.78 | 26.91 | 23.60 | |
| **Qwen3-0.6B** | 61.13 | | | | | | | 7.76 |
| Compression-Tokens (Causal) | | 48.09 | 46.30 | 34.68 | 36.05 | 20.10 | 20.69 | |
| Compression-Tokens (Bidirectional) | | 48.04 | 50.67 | 38.85 | 40.35 | 21.29 | 21.07 | |
| Mean-Pooling | | 56.48 | 53.12 | 42.47 | 42.21 | 21.29 | 19.06 | |
| **Gemma2-2B** | 66.47 | | | | | | | 10.34 |
| Compression-Tokens (Causal) | | 56.17 | 55.78 | 47.16 | 46.68 | 31.17 | 29.72 | |
| Compression-Tokens (Bidirectional) | | 59.20 | 59.51 | 48.59 | 51.43 | 31.50 | 31.91 | |
| Mean-Pooling | | 64.29 | 63.44 | 56.94 | 55.71 | 33.42 | 31.79 | |
| **Llama3.2-1B** | 61.67 | | | | | | | 9.03 |
| Compression-Tokens (Causal) | | 48.57 | 45.60 | 38.44 | 37.12 | 23.03 | 23.14 | |
| Compression-Tokens (Bidirectional) | | 52.24 | 49.98 | 40.65 | 42.08 | 23.59 | 23.72 | |
| Mean-Pooling | | 57.97 | 55.15 | 38.58 | 46.23 | 19.53 | 23.17 | |

Table 10: NarrativeQA $F_1$

### B.2.2 Out of Domain Datasets Results

| | Original | 4x | | 16x | | 128x | | No Ctx |
|---|---|---|---|---|---|---|---|---|
| | | Single | Multi | Single | Multi | Single | Multi | |
| **Baseline Systems** | | | | | | | | |
| *LLMLingua2* (Qwen3-8B) | | | 65.65 | | 46.46 | | 52.55 | |
| *ICAE* (Mistral-7B) | | 70.63 | | | | | | |
| *PCC Lite* (GPT2-Large & Llama3.1-8B) | | 86.43 | | 78.03 | | 72.50 | | |
| *PCC Large* (Llama3.1-8B) | | 86.64 | | 77.13 | | 74.08 | | |
| **Our Methods** | | | | | | | | |
| **Qwen3-8B** | 89.65 | | | | | | | 53.79 |
| Compression-Tokens (Causal) | | 89.67 | 89.15 | 88.92 | 85.50 | 79.36 | 77.55 | |
| Compression-Tokens (Bidirectional) | | 90.44 | 89.41 | 87.07 | 87.95 | 75.90 | 78.17 | |
| Mean-Pooling | | 87.94 | 86.52 | 84.38 | 86.32 | 79.74 | 75.89 | |
| **Qwen3-4B** | 90.46 | | | | | | | 43.49 |
| Compression-Tokens (Causal) | | 88.49 | 83.28 | 82.95 | 80.42 | 72.27 | 70.84 | |
| Compression-Tokens (Bidirectional) | | 91.59 | 90.83 | 86.10 | 87.53 | 67.43 | 74.52 | |
| Mean-Pooling | | 85.50 | 88.72 | 83.68 | 85.32 | 71.04 | 67.50 | |
| **Qwen3-1.7B** | 89.20 | | | | | | | 25.08 |
| Compression-Tokens (Causal) | | 74.89 | 83.83 | 80.55 | 73.91 | 61.72 | 64.02 | |
| Compression-Tokens (Bidirectional) | | 85.62 | 86.41 | 76.15 | 80.34 | 61.46 | 61.67 | |
| Mean-Pooling | | 85.83 | 82.75 | 80.61 | 75.94 | 63.03 | 53.77 | |
| **Qwen3-0.6B** | 81.55 | | | | | | | 9.87 |
| Compression-Tokens (Causal) | | 78.27 | 73.57 | 62.14 | 64.79 | 48.97 | 49.69 | |
| Compression-Tokens (Bidirectional) | | 81.58 | 78.38 | 69.22 | 74.16 | 50.24 | 54.05 | |
| Mean-Pooling | | 81.45 | 77.88 | 69.08 | 70.41 | 52.45 | 43.08 | |
| **Gemma2-2B** | 90.69 | | | | | | | 54.06 |
| Compression-Tokens (Causal) | | 89.75 | 85.06 | 82.73 | 79.19 | 77.64 | 73.71 | |
| Compression-Tokens (Bidirectional) | | 88.52 | 87.14 | 85.32 | 84.63 | 78.15 | 73.59 | |
| Mean-Pooling | | 89.09 | 86.99 | 84.95 | 85.27 | 75.30 | 74.29 | |
| **Llama3.2-1B** | 82.13 | | | | | | | 31.14 |
| Compression-Tokens (Causal) | | 84.40 | 79.82 | 75.47 | 74.28 | 65.03 | 67.58 | |
| Compression-Tokens (Bidirectional) | | 83.72 | 84.18 | 78.94 | 73.03 | 64.83 | 68.57 | |
| Mean-Pooling | | 84.87 | 83.62 | 73.95 | 76.02 | 62.02 | 60.28 | |

Table 11: TriviaQA Verified $F_1$.

| | Original | 4x | | 16x | | 128x | | No Ctx |
|---|---|---|---|---|---|---|---|---|
| | | **Single** | **Multi** | **Single** | **Multi** | **Single** | **Multi** | |
| **Baseline Systems** | | | | | | | | |
| *LLMLingua2* (Qwen3-8B) | | | 32.79 | | 19.56 | | 18.76 | |
| *ICAE* (Mistral-7B) | | 27.34 | | | | | | |
| *PCC Lite* (GPT2-Large & Llama3.1-8B) | | 42.51 | | 35.36 | | 26.44 | | |
| *PCC Large* (Llama3.1-8B) | | 44.09 | | 33.39 | | 27.52 | | |
| **Our Methods** | | | | | | | | |
| **Qwen3-8B** | 60.44 | | | | | | | 19.15 |
| Compression-Tokens (Causal) | | 47.26 | 45.97 | 36.35 | 39.16 | 32.42 | 31.15 | |
| Compression-Tokens (Bidirectional) | | 51.46 | 51.51 | 40.05 | 42.51 | 32.54 | 32.69 | |
| Mean-Pooling | | 53.71 | 53.32 | 42.55 | 45.07 | 32.52 | 31.36 | |
| **Qwen3-4B** | 57.04 | | | | | | | 17.38 |
| Compression-Tokens (Causal) | | 45.07 | 43.44 | 34.61 | 34.34 | 28.95 | 26.25 | |
| Compression-Tokens (Bidirectional) | | 45.35 | 47.93 | 36.32 | 38.23 | 28.17 | 27.88 | |
| Mean-Pooling | | 51.47 | 48.49 | 40.36 | 39.09 | 28.84 | 27.31 | |
| **Qwen3-1.7B** | 46.62 | | | | | | | 14.86 |
| Compression-Tokens (Causal) | | 30.09 | 34.01 | 29.29 | 29.47 | 22.27 | 22.82 | |
| Compression-Tokens (Bidirectional) | | 37.92 | 37.00 | 29.72 | 31.44 | 22.43 | 21.05 | |
| Mean-Pooling | | 42.14 | 39.78 | 32.33 | 32.46 | 22.01 | 22.18 | |
| **Qwen3-0.6B** | 39.04 | | | | | | | 10.97 |
| Compression-Tokens (Causal) | | 29.69 | 28.58 | 23.55 | 23.95 | 18.80 | 19.46 | |
| Compression-Tokens (Bidirectional) | | 29.16 | 33.06 | 25.47 | 26.99 | 19.60 | 18.80 | |
| Mean-Pooling | | 33.84 | 32.70 | 26.11 | 26.21 | 19.52 | 18.08 | |
| **Gemma2-2B** | 51.45 | | | | | | | 16.76 |
| Compression-Tokens (Causal) | | 39.83 | 40.80 | 33.93 | 35.23 | 28.83 | 29.06 | |
| Compression-Tokens (Bidirectional) | | 40.76 | 41.96 | 34.73 | 35.52 | 29.47 | 27.27 | |
| Mean-Pooling | | 45.61 | 44.88 | 37.14 | 36.88 | 28.70 | 28.94 | |
| **Llama3.2-1B** | 39.75 | | | | | | | 14.30 |
| Compression-Tokens (Causal) | | 30.58 | 29.42 | 26.27 | 26.66 | 21.04 | 21.63 | |
| Compression-Tokens (Bidirectional) | | 31.71 | 30.30 | 25.77 | 28.88 | 23.74 | 20.71 | |
| Mean-Pooling | | 34.84 | 33.60 | 27.18 | 27.29 | 21.21 | 20.46 | |

Table 12: AdversarialQA $F_1$

| | Original | 4x | | 16x | | 128x | | No Ctx |
|---|---|---|---|---|---|---|---|---|
| | | Single | Multi | Single | Multi | Single | Multi | |
| **Baseline Systems** | | | | | | | | |
| *LLMLingua2* (Qwen3-8B) | | | 27.42 | | 15.32 | | 7.56 | |
| *ICAE* (Mistral-7B) | | 28.42 | | | | | | |
| *PCC Lite* (GPT2-Large & Llama3.1-8B) | | 46.31 | | 33.25 | | 18.14 | | |
| *PCC Large* (Llama3.1-8B) | | 46.77 | | 31.17 | | 17.95 | | |
| **Our Methods** | | | | | | | | |
| **Qwen3-8B** | 56.77 | | | | | | | 7.49 |
| Compression-Tokens (Causal) | | 49.69 | 48.77 | 40.37 | 41.71 | 30.94 | 28.40 | |
| Compression-Tokens (Bidirectional) | | 51.65 | 51.68 | 44.91 | 45.85 | 31.40 | 31.07 | |
| Mean-Pooling | | 55.36 | 53.87 | 48.95 | 48.63 | 31.79 | 29.12 | |
| **Qwen3-4B** | 56.14 | | | | | | | 6.59 |
| Compression-Tokens (Causal) | | 47.85 | 46.50 | 40.73 | 40.49 | 28.86 | 27.16 | |
| Compression-Tokens (Bidirectional) | | 49.63 | 50.74 | 42.98 | 44.60 | 27.72 | 28.07 | |
| Mean-Pooling | | 54.74 | 53.24 | 46.93 | 47.01 | 29.95 | 26.25 | |
| **Qwen3-1.7B** | 54.75 | | | | | | | 5.09 |
| Compression-Tokens (Causal) | | 37.53 | 43.23 | 36.08 | 35.38 | 24.47 | 22.71 | |
| Compression-Tokens (Bidirectional) | | 46.33 | 47.36 | 39.28 | 39.72 | 24.81 | 23.81 | |
| Mean-Pooling | | 52.04 | 50.67 | 42.90 | 42.55 | 25.16 | 22.02 | |
| **Qwen3-0.6B** | 51.54 | | | | | | | 4.82 |
| Compression-Tokens (Causal) | | 42.04 | 39.38 | 32.91 | 31.36 | 20.07 | 19.73 | |
| Compression-Tokens (Bidirectional) | | 43.34 | 43.74 | 34.15 | 35.77 | 21.12 | 19.16 | |
| Mean-Pooling | | 48.26 | 46.29 | 38.13 | 37.81 | 21.24 | 17.03 | |
| **Gemma2-2B** | 56.00 | | | | | | | 7.10 |
| Compression-Tokens (Causal) | | 48.10 | 46.86 | 41.66 | 40.33 | 29.30 | 27.74 | |
| Compression-Tokens (Bidirectional) | | 49.36 | 49.83 | 42.30 | 43.72 | 29.85 | 28.81 | |
| Mean-Pooling | | 54.39 | 53.39 | 47.82 | 47.83 | 31.16 | 28.91 | |
| **Llama3.2-1B** | 52.26 | | | | | | | 5.94 |
| Compression-Tokens (Causal) | | 41.97 | 40.07 | 35.08 | 35.24 | 22.50 | 23.14 | |
| Compression-Tokens (Bidirectional) | | 44.58 | 44.02 | 36.36 | 38.77 | 23.84 | 23.88 | |
| Mean-Pooling | | 49.90 | 46.92 | 33.84 | 39.59 | 17.67 | 21.17 | |

Table 13: ParaphraseRC $F_1$

## C  LLM USAGE

LLMs (specifically, ChatGPT) were used in the process of writing this paper for creating tables and figures, as well as proof-reading.

