# OpenReview forum: "Simple Context Compression: Mean-Pooling and Multi-Ratio Training"
_ICLR.cc/2026/Conference — ICLR 2026 Conference Withdrawn Submission_

### Official Review · Reviewer_hM8F · 2025-10-28

**Soundness:** 3
**Presentation:** 3
**Contribution:** 3
**Rating:** 6
**Confidence:** 4

**Summary:**

This paper addresses the critical problem of computational and memory costs associated with using long contexts in RAG by proposing a simplified and highly efficient soft context compression approach. The authors introduce Mean-Pooling, a lightweight architecture that utilizes a standard encoder LLM followed by a non-overlapping mean-pooling operation to transform the long context into a shorter, continuous representation. Empirically, the Mean-Pooling method consistently outperforms the widely used compression-tokens architecture. Furthermore, the paper proposes a novel Multi-Ratio Training strategy, allowing a single compressor model to be trained efficiently to operate effectively across a broad range of compression ratios (e.g., 4x to 128x), enhancing deployment flexibility. The claims are supported by extensive experiments across multiple LLM families (Qwen3, Gemma2, Llama3.2) and various QA benchmarks.

**Strengths:**

1.  The core contribution, Mean-Pooling, is remarkably simple and elegant. By directly leveraging the hidden states of an encoder LLM and applying a non-overlapping mean-pooling, the method introduces zero additional parameters beyond the encoder, making it highly resource-efficient and easy to integrate compared to methods that rely on learnable tokens.
2.  The paper provides compelling evidence that this simple Mean-Pooling approach consistently outperforms the more complex compression-tokens baseline across diverse experimental setups (different models, sizes, and datasets), demonstrating its effectiveness and robustness.
3.  The unified training strategy for multiple compression ratios is a significant practical contribution. It solves the challenge of training and deploying separate models for varying computational budgets, showing that a single model can generalize across ratios with minimal performance degradation.
4.  The experiments are extensive, covering model scales from 0.6B to 8B and incorporating both in-domain and out-of-domain evaluation datasets. This broad empirical validation lends high credibility to the main findings.

**Weaknesses:**

1.  The Mean-Pooling approach aggregates information, leading to a less interpretable compressed representation compared to methods that explicitly select important tokens. The paper could benefit from an analysis of which types of information (e.g., critical facts, supporting details, filler text) are best preserved or lost under this averaging scheme.
2.  The primary cost saving is in the generation phase (reduced KV cache). However, the initial encoding still requires running the full-length context through the encoder LLM. A clearer and more direct comparison of the total end-to-end latency/cost (encoding + generation) against other context compression methods, especially token-selection or sparse attention techniques, would provide a more complete picture of the practical utility in a real-world RAG system.

**Questions:**

Please refer to weakness.

---

### Official Review · Reviewer_1uwS · 2025-10-31

**Soundness:** 2
**Presentation:** 2
**Contribution:** 2
**Rating:** 2
**Confidence:** 4

**Summary:**

This paper revisits soft context compression for long-document QA and proposes an ultra-simple alternative: mean-pooling the encoder’s final hidden states. Without extra parameters, the compressor is distilled once to serve multiple compression ratios (4×–128×). Across six QA datasets and model scales 0.6 B–8 B, mean-pooling consistently outperforms conventional compression-token baselines; multi-ratio training incurs only a minor drop while yielding a single, compute-budget-agnostic model.

**Strengths:**

1. The paper conducts some experiments.
2. Figures of paper are well presented.

**Weaknesses:**

1. It seems like the tables in this paper are not well organized. Some cells are blanked without giving any reason.
2. The baselines are run with different models (lines 327 - 330), which makes them not comparable.
3. Formatting problem, line 371-374 shows a large redundant black space (also the beginning of page 6).

**Questions:**

No, see weakness.

---

### Official Review · Reviewer_pWoR · 2025-10-31

**Soundness:** 2
**Presentation:** 3
**Contribution:** 1
**Rating:** 2
**Confidence:** 4

**Summary:**

This paper addresses the computational challenges of long-context Retrieval-Augmented Generation (RAG) by investigating soft context compression. The core idea is to pre-compute a shorter, continuous representation of a long document, thereby reducing the memory (KV cache) and time (self-attention) costs during generation.
The authors propose a simple and lightweight compressor architecture and explore the feasibility of training a single model to handle multiple compression ratios.
Contributions：
A Simple and Effective Compressor Architecture: The paper introduces a mean-pooling-based compressor that outperforms the widely used compression-tokens approach.
The proposed mean-pooling approach consistently achieves the strongest performance and is more efficient than the compression-tokens baseline.
A simple modification to the attention pattern in the compression-tokens method can significantly (but not entirely) close the performance gap.
Compression quality improves with model scale, suggesting greater benefits for larger models.

**Strengths:**

(1)  Comprehensive Experimental Setup
Evaluations span multiple datasets (e.g., SQuAD, HotpotQA, NarrativeQA), covering both single-hop and multi-hop reasoning tasks with diverse context lengths.
Performance is reported under various compression ratios (4×, 16×, 128×), enabling analysis of scalability and degradation patterns.
Inclusion of ablated baselines such as “No Ctx” helps isolate whether performance stems from actual context utilization or prior knowledge.
(2)  Good clarity
The paper excels in structural organization and readability.

**Weaknesses:**

(1)  Lack of originality
The proposed method constitutes an incremental advancement over the In-Context Autoencoder (ICAE) for context compression.
Its core contribution—a mean pooling operation for feature aggregation—is a straightforward and intuitive technique, but it lacks novelty.
Furthermore, the approach bears significant resemblance to other existing methods, such as those employing top-k selection for context compression, thereby further diminishing its originality.
(2)  Lack of Mechanistic Understanding: Why Does Mean-Pooling Work?
The paper presents mean-pooling as an effective compression operator but offers no analysis into why it works, especially compared to alternatives (e.g., max-pooling, attention-based weighting, Top-K).
This limits the contribution from being merely empirical ("it works well") to potentially explanatory ("here's why pooling suffices").
For example:
Is uniform averaging optimal because LLM encoders produce near-homogeneous representations in semantically coherent blocks?
Or does performance stem from the fact that retrieval systems often return redundant information, making averaging a noise-reduction mechanism?
Without such investigation, the method risks appearing heuristic rather than principled.
(3)  Ambiguity in Training-Deployment Mismatch for Multi-Ratio Setup
The multi-ratio training strategy trains a single model on variable compression levels (4×, 16×, etc.), enabling flexible deployment. However, the paper lacks clarity on how ratio selection occurs during training:
Are all ratios used in every batch (balanced sampling)?
Or is one ratio sampled per instance? If so, what distribution?
During inference, can the model handle unseen ratios (e.g., trained on 4×/16×, tested on 8×)? No evidence is provided.
This undermines claims of “flexible deployment,” as the robustness of interpolation between trained ratios remains unverified.

**Questions:**

Refer to the weaknesses.

---

### Official Review · Reviewer_EvS9 · 2025-11-01

**Soundness:** 3
**Presentation:** 3
**Contribution:** 2
**Rating:** 2
**Confidence:** 4

**Summary:**

This paper studies *soft context compression* for large language models. It introduces two key ideas:

1. Mean-Pooling Compressor: a non-parametric, lightweight architecture that encodes the input with a Transformer encoder and performs non-overlapping mean pooling (stride = compression ratio r) followed by a linear projection to produce continuous compressed representations.
2. Multi-Ratio Training: a unified distillation regime where a single compressor is trained to handle multiple compression ratios simultaneously by matching the teacher model’s token-level distributions.

Experiments across multiple model families and sizes (Llama 3.2, Gemma 2, Qwen 3; 0.6 B – 8 B parameters) and 6 QA datasets show that mean-pooling outperforms token-based compressors (causal and bidirectional). Multi-ratio training enables a single model to operate across ratios 4× – 128× with only minor degradation, though gains are not uniform across methods.

**Strengths:**

- Simplicity and efficiency: The mean-pooling compressor achieves strong performance with no additional tokens or complex modules. It consistently outperforms causal and bidirectional token compressors across models and ratios.

- Comprehensive evaluation: Experiments cover multiple families/scales and both single- and multi-ratio settings, with clear teacher-normalized metrics. (though the experiment setting is not based on the real RAG where the noisy information may dominate the context)
- Rigorous ablations: Freezing/removing encoder/decoder, omitting the linear layer, and random ratio sampling quantify where performance originates.

**Weaknesses:**

- Limited context length: All experiments cap sequences at 1024 tokens, far below realistic long-context (e.g., 32 k – 128 k) settings where compression is most needed.
- Minimal benefit from multi-ratio training: For mean-pooling, multi-ratio training yields similar or slightly worse scores than single-ratio training, implying limited synergy between the two contributions. In contrast, bidirectional token compressors benefit more.
- Also, the two ideas—mean-pooling and multi-ratio training—address orthogonal aspects (architecture vs. training). Their connection is mainly conceptual (“supporting one compressor across budgets”) rather than algorithmically coupled.

- Baseline coverage: The paper omits direct comparison to 500xCompressor (a recent improvement over Compression-Tokens approach).

reference
500xCompressor: Generalized Prompt Compression for Large Language Models

**Questions:**

- What is the core limitation of  soft-token based compression and why mean pooling makes a signification difference?

---

### Note · Authors · 2025-11-18

**Comment:**

We thank the committee for its work. The quality of reviews and concerns that are not aligned with experimental norms indicate there is no realistic avenue for discussion. We therefore withdraw the paper.

**Withdrawal Confirmation:**

I have read and agree with the venue's withdrawal policy on behalf of myself and my co-authors.